# Weyl spin-momentum locking in a chiral topological semimetal

Jonas A. Krieger [1,15,16], Samuel Stolz[2,3,16], Iñigo Robredo[4,5,16], Kaustuv Manna [6], Emily C. McFarlane[1], Mihir Date [1], Banabir Pal[1], Jiabao Yang[1], Eduardo B. Guedes[7,8], J. Hugo Dil [7,8], Craig M. Polley[9], Mats Leandersson[9], Chandra Shekhar [4], Horst Borrmann[4], Qun Yang[4], Mao Lin[10], Vladimir N. Strocov [7], Marco Caputo[7], Matthew D. Watson [11], Timur K. Kim [11], Cephise Cacho [11], Federico Mazzola [12,13], Jun Fujii [14], Ivana Vobornik [14], Stuart S. P. Parkin [1], Barry Bradlyn [10], Claudia Felser [4], Maia G. Vergniory [4,5] & Niels B. M. Schröter [1] ✉

Spin-orbit coupling in noncentrosymmetric crystals leads to spin-momentum locking – a directional relationship between an electron's spin angular momentum and its linear momentum. Isotropic orthogonal Rashba spin-momentum locking has been studied for decades, while its counterpart, isotropic parallel Weyl spin-momentum locking has remained elusive in experiments. Theory predicts that Weyl spin-momentum locking can only be realized in structurally chiral cubic crystals in the vicinity of Kramers-Weyl or multifold fermions. Here, we use spin- and angle-resolved photoemission spectroscopy to evidence Weyl spin-momentum locking of multifold fermions in the chiral topological semimetal PtGa. We find that the electron spin of the Fermi arc surface states is orthogonal to their Fermi surface contour for momenta close to the projection of the bulk multifold fermion at the Γ point, which is consistent with Weyl spin-momentum locking of the latter. The direct measurement of the bulk spin texture of the multifold fermion at the R point also displays Weyl spin-momentum locking. The discovery of Weyl spin-momentum locking may lead to energy-efficient memory devices and Josephson diodes based on chiral topological semimetals.

Objects that lack inversion and mirror symmetries are chiral and often show an intimate relationship between the direction of their spin angular momentum and their linear momentum. For instance, in particle physics, the chirality of the hypothetical massless Weyl-fermion determines if its spin is either parallel or antiparallel to its momentum direction[1]. In chemistry and biology, chiral molecules such as DNA have been shown to act as effective spin polarisers, via an effect known as chirality-induced-spin-selectivity (CISS)[2,3]. CISS has recently been attributed to a locking of the linear electron momentum to the electron's orbital angular momentum, which is linked to the spin via spin–orbit coupling (SOC)[4]. In condensed matter physics, three

fundamental prototypes of spin–momentum locking (SML) can be distinguished to leading order in a $\mathbf{k} \cdot \mathbf{p}$ expansion (cf. Fig. 1a): Rashba SML[5], where the spin is isotropically locked orthogonal to the electron's linear momentum; Dresselhaus SML[6], where the locking is mixed orthogonal-parallel, depending on the momentum direction; and Weyl SML, where the spin and momentum are parallel along all momentum directions. Whilst Rashba and Dresselhaus SML can occur in achiral noncentrosymmetric crystals and have been studied for decades[7], Weyl SML is only allowed in chiral cubic crystals and has so far remained elusive in experiments. Note that generic Weyl-fermions, which have been found in many achiral Weyl semimetals, are typically

described by a Hamiltonian where the Pauli matrices act on an orbital pseudospin degree of freedom and not the physical electron spin[8].

It has been argued that generic structurally chiral crystals can exhibit Weyl SML due to the absence of mirror planes. However, without cubic crystalline symmetries, they typically have the spin locked parallel to the momentum only along one dimension or a special rotational axis, but not along all three dimensions simultaneously, as in the case of Weyl SML. An example of this is trigonal semiconducting tellurium, where the spin is locked parallel to the momentum only along a single screw axis[9–12].

Finding a material with true Weyl SML would not only complete the trio of prototypical forms of linear spin–momentum locking, but could also enable novel applications that are impossible to realize with any other spin texture: running a charge current through such material would produce a parallel spin-current whose spin direction is parallel to both charge and spin currents, which could find applications in energy-efficient memory devices[13,14]. In non-cubic chiral crystal such a longitudinal coupling between magnetization and current has been observed along specific directions[15,16]. In contrast, in the presence of Weyl SML this behavior should be independent of the crystallographic direction along which the current is flowing. It is therefore particularly advantageous for thin film devices, where rotational grain disorder is common and thus precludes the alignment of the current direction with a special crystallographic axis. Moreover, interfaces between chiral topological semimetals and magnetic and superconducting overlayers can be fabricated on any of the many possible surface planes of the cubic chiral crystals, which could facilitate lattice matching for device applications. Moreover, we predict that such materials could also display a new type of parallel Josephson diode effect[17], which could lead to novel superconducting circuit elements[18] (see Supplementary Note 9 for more details).

Isotropic parallel Weyl SML has recently been predicted for chiral cubic topological semimetals[8,13], a class of quantum materials that host singly or multiply degenerate nodal electronic band crossings with large topological charges[19–24]. The screw and rotational axes of cubic point groups {T,O} force the effective Hamiltonian describing such singly degenerate crossings to take the form of the isotropic Weyl-Hamiltonian $H \propto \mathbf{k} \cdot \boldsymbol{\sigma}$, where $\mathbf{k}$ is the crystal momentum wave vector and $\boldsymbol{\sigma}$ is a vector of Pauli matrices acting on the physical electron spin.

Such quasiparticles would therefore exhibit true three-dimensional isotropic Weyl-type parallel spin–momentum locking.

We recently developed a theory showing that Fermi surfaces formed by multifold fermions also possess spin–momentum locking close to the degeneracy point[25]. Whilst various complex spin textures are possible depending on the involved orbitals, the simplest possible spin texture shows, as in Kramers–Weyl fermions, isotropic parallel spin–momentum locking. Additionally, Refs. 11,26 considered the spin texture in chiral crystals for bands that can be treated in the limit of weak SOC, and derived the possible spin textures for weakly spin-split Fermi surfaces, finding many cases where chiral cubic symmetry enforces perfectly isotropic parallel SML. In the context of multifold fermions, Refs. 11,26 apply to Fermi pockets far enough from the nodal point such that bands are only weakly spin-split; this complements the approach of Ref. 25 which is applicable much closer to the nodal point, where bands cannot be described in a simple spin–orbit decoupled basis.

Multifold fermions that could realize Weyl SML are the fourfold degenerate spin $S = 3/2$ Rarita–Schwinger–Weyl multifold fermion and the sixfold degenerate $S = 1$ multifold fermion, which were predicted to occur in materials of the cubic B20 crystal structure (space group 198) at the Γ and R points, respectively[19,27–29]. Recent experiments confirmed the existence of the multifold fermions in several B20 materials, such as CoSi[20,21], RhSi[22], PdGa[24], AlPt[23], or PtGa[30,31] by directly imaging the nodal crossings and determining the Chern number by counting the Fermi-arc surface states with angle-resolved photoemission spectroscopy (ARPES). Moreover, a direct link between the sign of the Chern numbers and the handedness of the host crystal has been established[24,32]. However, this prior work did not study the exotic spin textures of these multifold fermions. Therefore, experimental evidence for Weyl SML still remains elusive until now.

The method of choice to study momentum-resolved spin textures is spin-resolved ARPES (spin-ARPES). However, spin-ARPES measurements of Kramers–Weyl and multifold fermions are challenging for two reasons: Firstly, accurate measurements of bulk band structures around 3D nodal points typically require soft X-ray ARPES to obtain a high out-of-plane momentum resolution. Because of the low photoemission cross-section in the soft X-ray energy range, spin-resolved soft X-ray ARPES with the currently available single-channel spin-

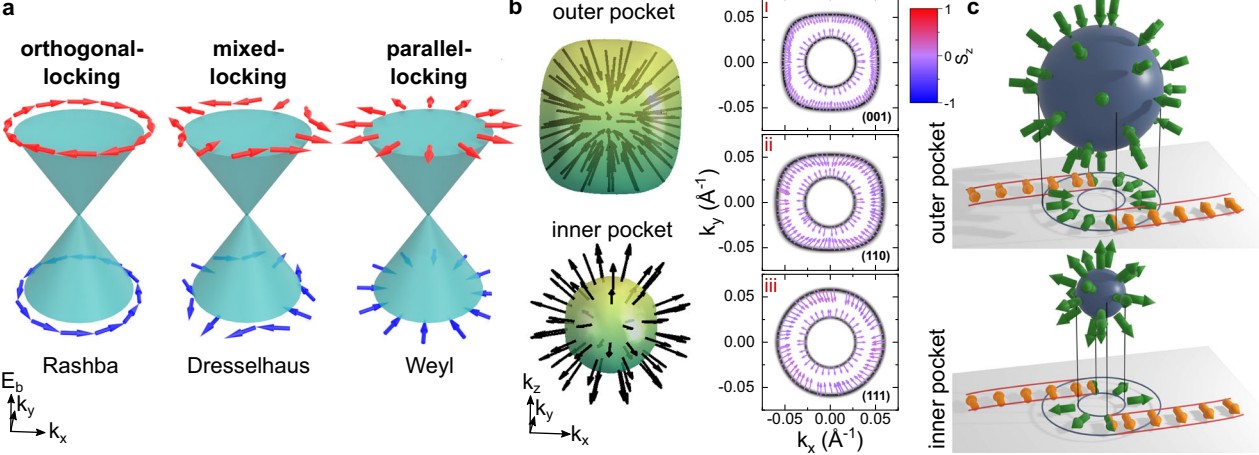

**Fig. 1 | Spin–momentum locking prototypes and spin texture of Rarita–Schwinger–Weyl fermion in PtGa. a** Overview of the prototypical forms of spin–momentum locking (SML) linear in momentum, (left) orthogonal isotropic Rashba SML with Hamiltonian $H = k_x\sigma_y - k_y\sigma_x$, (middle) mixed orthogonal-parallel Dresselhaus SML $H = k_x\sigma_x - k_y\sigma_y$, and (right) isotropic parallel Weyl SML $H = k_x\sigma_x + k_y\sigma_y$ for $k_z = 0$. **b** Calculated three-dimensional spin texture of the two spherical Fermi surfaces formed by the $S = 3/2$ Rarita–Schwinger–Weyl multifold fermion in PtGa. Both electron pockets are centred at the Γ point. Panels (i-iii) show two-dimensional slices through the three-dimensional Fermi surface, indicating the radial nature of the spin texture along all momentum directions. $S_z$ denotes the out-of-plane component of the calculated spin polarization and the corresponding color coding applies to all three panels. **c** Illustration of the link between the spin texture of the bulk multifold fermions (green arrows) and the spin texture of the topological Fermi-arcs surface states (orange arrows). Since the Fermi arcs inherit the spin texture of the bulk bands in the $k_z = 0$ plane, their spin texture is orthogonal to their Fermi surface contour.

detectors is typically unfeasible due to low count rates. Secondly, in many materials, the separation between Kramers−Weyl or multifold fermions and other spin-polarized bands is small in energy and momentum, which makes it difficult to disentangle the contributions of all individual bands due to the limited resolution. For instance, for the material PtGa, previous studies reported the spin-polarization of the bulk states consistent with time-reversal symmetry[31], but no signatures of spin−momentum locking.

## Results

In the present work, we overcome these obstacles by inferring the Weyl SML of the bulk multifold fermions in PtGa from an observation of the Fermi-arc spin texture, and by direct measurement of the bulk spin texture, which is consistent with our expectation from our theoretical models. Our ab initio calculations predict that for PtGa the multifold fermion at the Γ point is located just below the Fermi level and forms two electron pockets on the Fermi surface (Fig. 1b): the inner one is practically spherical and displays Weyl SML. The outer pocket is almost spherical, with small corrections towards a rectangular shape that lead to minor deviations from parallel SML. Since the bands making up the multifold fermion at Γ are only weakly split by SOC (see Supplementary Fig. 5 for a band structure calculation without SOC), parallel SML is also consistent with the prediction of Refs. 11,26 for B20 crystal structure.

Turning to the surface states, we note that the Fermi arcs connect and simultaneously are derived from the surface-projected Fermi surface pockets of multifold fermions with opposite Chern number[33]. In the absence of SOC, two spin-degenerate Fermi arcs will emanate from the bulk projection of the Fermi surface. Our theoretical analysis in Supplementary Note 7 considers the leading-order spin-orbit correction where the multifold crossing is well described by

$$
\begin{aligned}
H(\mathbf{k}) \approx & \, v_{\mathrm{F}} \mathbf{k} \cdot \mathbf{L} + \lambda_0 \mathbf{L} \cdot \boldsymbol{\sigma} \\
& + \lambda_1 \mathbf{k} \cdot \boldsymbol{\sigma} + \lambda_2 \mathbf{k} \cdot (\tilde{\boldsymbol{L}} \times \boldsymbol{\sigma}) + \lambda_3 \Lambda_{ijk} k_i \tilde{L}_j \sigma_k,
\end{aligned}
\tag{1}
$$

where $\mathbf{k}$ describes the linear momentum, $\boldsymbol{\sigma}$ are Pauli matrices acting on the physical spin, $\mathbf{L}$ is a vector of higher-order pseudospin matrices, $\tilde{\mathbf{L}}$ is a vector of time-reversal even symmetric matrices[19,34] (see Supplementary Note 7), $v_{\mathrm{F}}$ is the Fermi velocity, $\lambda_{\mathrm{O}}$ is the magnitude of the $k$-independent bulk spin-orbit interaction near Γ and $\lambda_i$ are the $k$-dependent spin−orbit interaction terms. By fitting this model to our DFT calculations of PtGa, we find that for the bands at the Fermi-level near the Γ point, we are in the limit of $\lambda_i |\mathbf{k}| < \lambda_{\mathrm{O}} < v_{\mathrm{F}}|\mathbf{k}|$. In this regime, if we are focused on the highest energy bands of $H_{\mathrm{O}}$ we can neglect $\lambda_i (i = 1, 2, 3)$ and treat $\lambda_{\mathrm{O}}$ as a perturbation to $H_{\mathrm{O}} = v_{\mathrm{F}} \mathbf{k} \cdot \mathbf{L}$. In this approximation, we find that SOC splits the bulk bands with a Weyl-type SML. Additionally, we find that the Fermi arc surface states are given by the highest pseudospin eigenstates of $\hat{\mathbf{n}} \cdot \mathbf{L}$ where $\hat{\mathbf{n}}$ is the unit vector normal to the arc direction (and with zero $z$ component). To first order in perturbation theory (and in the limit of $\lambda_i \to 0$), SOC splits the spin degeneracy of the Fermi arcs with an energy splitting $\Delta H(\mathbf{k}) = \lambda_0 \hat{\mathbf{n}} \cdot \boldsymbol{\sigma}$. Thus, the bulk parallel SML at the Fermi surface leads to the locking of the Fermi arc spin orthogonal to its Fermi surface contour (i.e., the spin is parallel to the group velocity of the surface states). We can therefore see that the Fermi-arc surface states in the vicinity of the multifold node inherit the Weyl SML of the $k_z = 0$ bulk states as illustrated in Fig. 1c, since the group velocity of a Fermi arc tangentially merging with the bulk pocket is pointing along the same direction as the projected bulk momentum. Note that this analysis is only valid close to the degeneracy point. As we get away, the description in Eq. (1) breaks down, introducing higher-order terms that make the spin texture deviate from isotropic parallel spin−momentum locking.

By measuring the surface spin texture of the Fermi arcs sufficiently close in momentum to the projection of the multifold node, we can therefore infer the spin texture of the latter. For larger momenta

away from the multifold fermion, the $\mathbf{k} \cdot \mathbf{p}$ approximation will break down and the Fermi-arc spin texture will no longer be linked to the spin texture of the multifold fermions near the node. Because the Fermi arcs are non-dispersive in the out-of-plane direction, spin-ARPES can be measured at relatively low photon energies with high cross-section, which makes such experiments feasible with currently available synchrotron set-ups. Furthermore, since there are sufficiently large projected bulk band gaps at the Fermi surface of PtGa close to the multifold fermion at the Γ point, we can measure the Fermi-arc spin texture without interference from other bands.

The samples studied in our work were enantiopure PtGa single-crystals that were cut along different crystallographic directions, then polished and cleaned in vacuum by cycles of Ar-ion sputtering and annealing (see Methods section for more details). Our bulk sensitive soft X-ray ARPES measurements on PtGa crystals cut along the [111]-direction directly image the $S = 1$ multifold band crossing at the R point (Fig. 2a–b) and the $S = 3/2$ Rarita−Schwinger−Weyl multifold fermion at the Γ point (Fig. 2c–d), which are well reproduced by our ab-initio calculations. Note that the Rarita−Schwinger−Weyl fermion at the Γ point is only about 100 meV below the Fermi level, which means that the Fermi wave vector $\mathbf{k}_{\mathrm{F}}$ is close to the Γ point and thus the SML derived from the $\mathbf{k} \cdot \mathbf{p}$ approximation can be expected to hold on the Fermi surface.

With our more surface-sensitive VUV-ARPES measurements of the (001) surface of PtGa (Fig. 2e–h) we resolve two Fermi arcs close to the center of the surface Brillouin zone that have a Fermi surface contour approximately parallel to the $k_x$ axis, and which merge into the small projected bulk pockets of the multifold fermion at the Γ point (upper red arrow in Fig. 2e). Further away from the Γ point, these Fermi arcs wind from the centre of the surface Brillouin zone around the projected bulk pocket centered at the R point and ultimately merge with the latter. The Fermi arcs are well reproduced by our ab-initio calculations (Fig. 2f) and our photon energy dependent VUV-ARPES measurement confirms experimentally that they have no out-of-plane dispersion (Fig. 2g), as expected for 2D surface states. The experimental band dispersion along the $\bar{\Gamma}$-$\bar{R}$ high-symmetry direction is displayed in Fig. 2h, and shows how four Fermi arcs merge with the projection of the multifold fermion at the $\bar{\Gamma}$ point in the center of the surface Brillouin zone.

The resolution of the four individual Fermi arcs close to the multifold fermion at $\bar{\Gamma}$, which are well separated from other bands due to a large projected bulk band gap, enables us to measure their spin texture (Fig. 3). As illustrated in Fig. 3a, we find that close to $\bar{\Gamma}$, the spin polarization of the outer Fermi arc points anti-parallel to the $k_y$ axis, whilst the inner Fermi arc points parallel to the $k_y$ axis. For both Fermi arcs, the spins point orthogonal to the Fermi contour, which is also reproduced by our ab initio calculations (Fig. 3b). Since the Fermi arcs inherit the spin texture of the multifold fermion for momenta close to $\bar{\Gamma}$, we can infer from the spin texture of the Fermi arcs that the inner (outer) bulk pocket of the multifold fermion has a spin texture that is parallel (antiparallel) to the momentum pointing away from (towards) the Γ point. This is consistent with parallel Weyl-type parallel spin−momentum locking of the multifold fermion predicted by our bulk ab initio calculations.

We have also measured the spin texture of the Fermi arc in-between $\bar{\Gamma}$ and $\bar{R}$ (shown in Fig. 3c–e), expecting that there would be no spin−momentum locking due to a breakdown of the $\mathbf{k} \cdot \mathbf{p}$ approximation. Surprisingly, we find that both the measured and calculated in-plane spin textures are predominantly tangential to the Fermi contour. This suggests that in this momentum region, the Rashba-effect due to surface inversion symmetry breaking could dominate.

To check that the measured spin texture of the Fermi arcs near $\bar{\Gamma}$ is indeed a property of the initial states and not influenced by matrix element effects (see Refs. 35−37), we have repeated the measurement with different light polarizations (s-, p-, right-circular and left-circular)

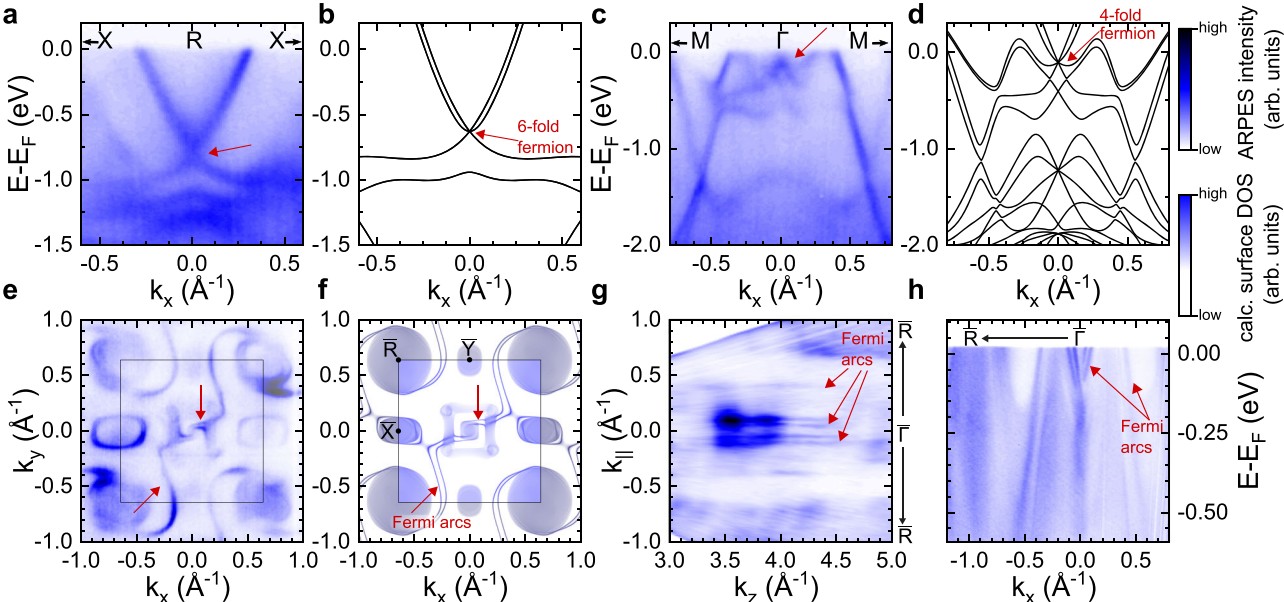

**Fig. 2 | Spin-integrated band structure of PtGa. a** Bulk band dispersion along the X - R - X direction measured with soft X-ray ARPES $hv = 547$ eV and $p$-polarization. The red arrow indicates the 6-fold band crossing located at the R point, which is well-reproduced by ab-initio calculations displayed in **b**. **c** Bulk band dispersion along the M - Γ - M direction measured with $hv = 653$ eV and $p$-polarization. The red arrow indicates the 4-fold Rarita--Schwinger--Weyl multifold fermion located at the Γ point that is located about 100 meV below the Fermi level, which is well-reproduced by our ab-initio calculations (**d**). **e** Fermi surface measured with surface

sensitive VUV-ARPES $hv = 60$ eV and $p$-polarization. Red arrows indicate the Fermi arcs close to the centre and at the boundary of the Brillouin zone. **f** Ab-initio surface state calculations of PtGa on the (100) surface. **g** Fermi surface along the in-plane $k_{\parallel}$ vs. out-of-plane $k_z$ momentum direction measured with photon energy dependent VUV-ARPES. $k_{\parallel}$ points along the diagonal of the surface Brillouin zone. **h** Band dispersion along the $\overline{\Gamma}$-$\overline{R}$ high symmetry direction, measured with $hv = 60$ eV and $p$-polarization.

and with a different experimental geometry at a different synchrotron beamline. We find that the extracted spin-polarization is practically identical for all measured polarizations (Fig. 4) and photon energies (see Supplementary Fig. 3), which is strong evidence that the observed spin texture reflects the initial equilibrium Bloch state with Weyl-type SML. We note that in the measurement $s$-polarized light Fig. 4 the surface state spectral weight is suppressed due to selection rules, preventing an accurate determination of the spin direction for this specific spectrum since the variation of the spin polarization of the surface state is small compared to the noise on the background contribution. As another consistency check, we have also investigated the spin texture of the Fermi arc of the opposite enantiomer and found that the spin texture behaves as expected from the transformational properties of a pseudovector (see Supplementary Fig. 4).

Due to the presence of $k_z$ broadening, most of the bulk bands near the multifold crossings in PtGa are insufficiently separated from each other to allow a well-defined spin-ARPES measurement with VUV energies. The notable exception is the lower branch of the multifold fermion at the R point (Fig. 2a, b), which as shown in Fig. 5a is still discernible as a single band, even when measured at $hv = 74$ eV, corresponding to $k_z = 7\pi/c$ in the free-electron final state model. To determine the appropriate photon energy to probe the R point, we have used the inner potential $V_0 = 12$ eV as determined in Ref. 38 for PtGa. The similarity between the bulk-sensitive SX-ARPES measurements shown in Fig. 2a and the spectrum measured at hv = 74 eV (Fig. 5a) further supports the correct choice of photon energies. To measure the spin-polarization of this band we have acquired energy distribution curves (EDCs) at different locations around the R point in the $k_z = 0$ plane, where the $k_z$ broadening is reduced because the Fermi surface changes only slowly along the $k_z$ direction. Due to the crystal symmetry the obtained results are representative of equivalent points in the $k_x = 0$ and $k_y = 0$ planes as well. The positions of these cuts are indicated with Roman numerals on a constant energy surface in Fig. 5b, which also shows the calculated spin texture from ab initio models. Here the spin-direction is indicated

by yellow arrows and shows a Weyl-type parallel spin−momentum locking. To investigate potential matrix element effects, we obtained spin-resolved measurements with four different light polarizations (Fig. 5c). The direction of the spin polarization of the band of interest within the energy window $E_{(d)}$ was extracted with respect to the background in the nearby projected bulk band gap $E_{ref}$, and is shown in Fig. 5d. The systematic error of this approach was estimated by moving $E_{ref}$ within the grey shaded region of the background in Fig. 5a, as described in more detail in the Supplementary Note 3. We note that the vast majority of the data points shown in Fig. 5d are close to the black dashed lines, consistent with Weyl-type parallel SML locking of the bulk band near the multifold crossing at R. Two outliers are the measurement with $C^-$-polarized light in cut iii, and the one with $s$-polarized light in cut ii. The former has almost the same degree of polarization as the background, hampering an accurate extraction of the spin direction. The latter could be related to spin-interference effects in the photoemission process[37]. However, the fact that the vast majority of the extracted spin directions point parallel to the momentum, also for the directions (i, iii) that are not along a high symmetry axis, provides strong evidence for Weyl-type isotropic parallel spin−momentum locking of the band forming the multifold fermion at the R point.

## Discussion

In conclusion, our combined spin-integrated and spin-resolved ARPES measurements in combination with our theoretical models provide strong evidence that the $S = 1$ multifold band crossing at the R point and the $S = 3/2$ Rarita−Schwinger−Weyl multifold fermion at the Γ point in PtGa display Weyl SOC with almost isotropic parallel spin−momentum locking. This finding makes chiral topological semimetals interesting materials to realize novel memory devices. By combining such a chiral material with a ferromagnet with perpendicular magnetic anisotropy (PMA) in a bilayer, one could deterministically switch the magnetization direction of the magnet by running a charge current orthogonal to the interface, without the need for magnetic fields that are commonly used

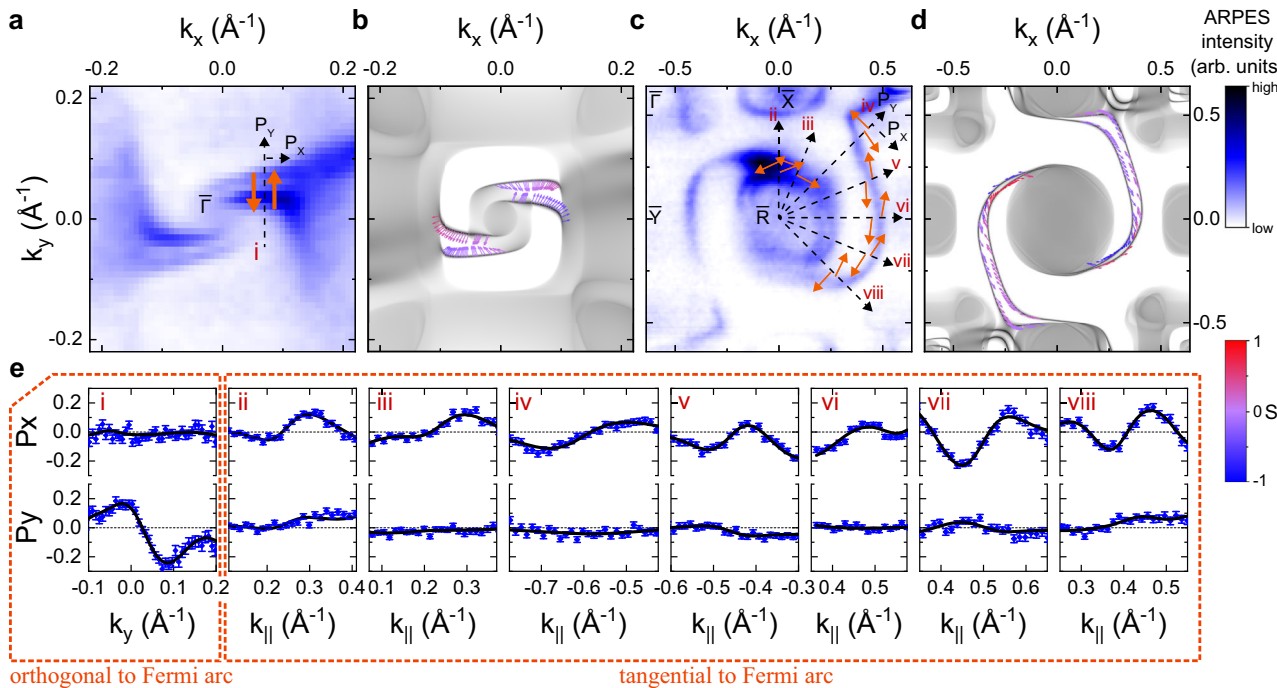

**Fig. 3 | Spin texture of the Fermi arcs on the (001) surface of PtGa. a** Spin texture of the two Fermi arcs close to the projection of the multifold fermion at the $\overline{\Gamma}$ point, orange arrows indicate the direction of the in-plane spin-polarization. **b** Calculation of the spin texture of the Fermi arcs around the $\overline{\Gamma}$ point, in good agreement with the experimental result. **c** Spin texture of the two Fermi arcs dispersing in-between $\overline{\Gamma}$ and $\overline{R}$ points. The orange arrows indicate the direction of the in-plane spin-

polarization. **d** Calculation of the Fermi-arc spin texture in-between $\overline{\Gamma}$ and $\overline{R}$. **e** Asymmetries in the measured in-plane spin-polarizations along various momentum directions indicated by the Roman numerals in **a**, **c**. The direction of positive $P_x$ and $P_y$ polarization is indicated with arrows on the Fermi surfaces in **a**, **c**. The black lines show fits to the data and the error bars indicate the statistical error due to counting statistics, see e.g., Ref. [44].

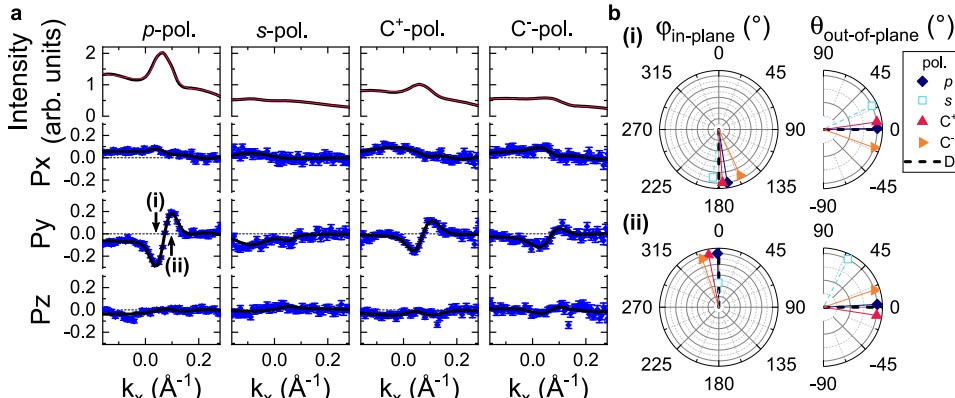

**Fig. 4 | Light polarization dependence of surface state spin texture near Γ.**
**a** Spin cuts corresponding to Fig. 3a measured with linear *s*- and *p*- polarized light and circular polarized light $C^{\pm}$ of opposite helicity. The black lines show fits to the data and the error bars indicate the statistical error due to counting statistics.
**b** Direction of the resulting in-plane ($\phi_{\text{in-plane}}$) and out-of-plane ($\theta_{\text{out-of-plane}}$) surface state polarization components for the two surface states labeled with (i) and (ii)

from **a**. Note that the extracted direction of *s*-pol. is unreliable, since the surface state is suppressed due to photoemission selection rules and thus the variation of the polarization across the surface state is within the statistical uncertainty. The black dashed lines correspond to the expected surface state spin-polarization orthogonal to the Fermi surface contour of the arc, c.f., Fig. 1c.

for PMA switching with the Rashba-effect or spin-Hall effect[13]. In contrast to conventional magnetic tunnel junctions used for field-free switching of PMA magnets, such a device would not require a tunnel barrier, thus potentially allowing for larger switching current densities. It could potentially also be more energy-efficient[13]. It has recently been predicted that the magnetization state of the PMA magnet could also be read out via magnetoresistance in such a bilayer[14], analogous to the unidirectional magnetoresistance reported in Ref. [39]. Such magnetoresistance could enable a memory device with only two terminals, thus enhancing

scalability compared to commonly employed three-terminal devices based on the Rashba- or spin-Hall effect. Moreover, since the magnetoelectric response in materials with Weyl-type SML is isotropic, device design is simplified because no preferred crystallographic alignment between the magnet and chiral semimetal is required. Furthermore, even in the presence of rotational disorder of crystal grains in a sample, the isotropic nature of Weyl SOC ensures that the magnetoelectric response of all grains adds up constructively. This advantage is particularly relevant for thin-film heterostructures in devices where grain

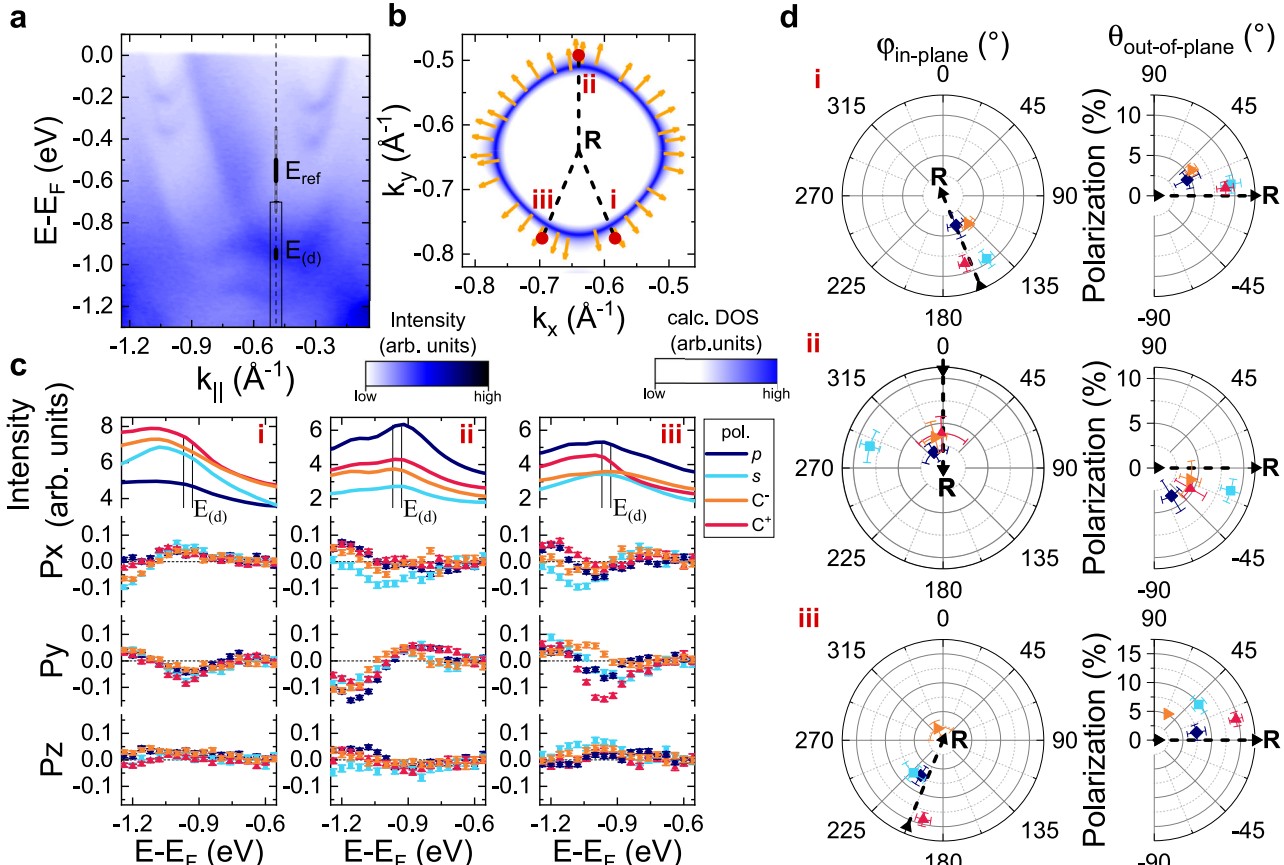

**Fig. 5 | Bulk spin texture of multifold fermion at R. a** Band dispersion along the R-$\overline{Y}$ high symmetry direction, measured with $h\nu = 74$ eV. **b** Calculated spin texture (yellow arrows) at $E_b = -0.75$ eV; corresponding to the band $E_{(d)}$ indicated in **a**. **c** Spin polarization as a function of binding energy at different momenta around R indicated by red dots and labelled by the Roman numerals in **b**. Different colors indicate different light polarizations. The position in $k$-space and the binding energy corresponding to cut ii are indicated in **a** by a dashed line and a black box, respectively. The spin polarization was assumed to be zero within the bulk band gap indicated by the black shaded region $E_{ref}$ in **a**. The error bars show the statistical error due to counting statistics. **d** In-plane ($\phi_{in\text{-}plane}$) and out-of-plane ($\theta_{out\text{-}of\text{-}plane}$) direction and magnitude of the spin polarization within the energy window $E_{(d)}$, shaded black and indicted by black lines in **a** and **c**, respectively. The dashed black line in **b** and **d** indicate the direction expected for Weyl-type SML around the R point. The error bars show a combination of statistical and systematical errors. The systematical error was estimated by varying the position of $E_{ref}$ within the gray shaded region in **a**. The precise calculation is provided in Supplementary Note 3.

disorder is common. Furthermore, Weyl-type SML could also enable Josephson diode effects[17] where the supercurrent tunnelling direction is parallel to a magnetic field or magnetization, which could be useful for low-dissipation readout of magnetic domains in cryogenic memory devices. By combining a Weyl-SML with a Rashba-SML material, one could potentially also tune the optimal angle between supercurrent and field (see Supplementary Note 9). The discovery of Weyl SML of multifold fermions in PtGa suggests that many other chiral cubic topological crystals hosting multifold fermions could display similar spin textures. For instance, many metallic members of the B20 crystal structure type are predicted or have been shown to host multifold fermions near the Fermi level. Therefore, numerous materials are available to test the exciting predictions for novel phenomena enabled by this new form of spin-orbit coupling.

## Methods

### Crystal growth

The growth and characterization method of the PtGa single crystals used in this study was previously published in Ref. 30. The single crystal samples were grown with a self-flux technique. Stochiometric and polycrystalline samples were first produced by arc melting and then crushed and then melted at 1150°C for 10 h. Subsequently, the sample was cooled to 1050°C at a rate of 1°C/h followed by subsequent cooling to 850°C at a rate of 50°C/h. Finally, the sample was annealed at 850°C for 120 h prior to being cooled to 500°C at a rate of 5°C/h, which further improved the sample quality. A detailed single crystallinity analysis with crystal orientation and a stoichiometry check were performed using Laue backscattering and energy-dispersive X-ray (EDX) spectroscopy, respectively. Homochirality was verified via Flack parameter analysis in a single-crystal X-ray diffractometer. Polished surfaces along different high symmetry directions were produced, after aligning the crystals at room temperature with a white-beam backscattering Laue X-ray setup.

### ARPES

Prior to any ARPES experiments, the PtGa single-crystals were cleaned in an ultrahigh vacuum with a base pressure better than $5 \times 10^{-9}$ mbar by repeated cycles of sputtering (Ar$^+$, 1 keV, -1 $\times 10^{-5}$ mbar) and annealing (970 K) until a clear LEED pattern was obtained.

Soft X-ray (SX-ARPES) experiments were performed at the SX-ARPES endstation[40] of the ADRESS beamline[41] at the Swiss Light Source, Switzerland. This endstation is equipped with a SPECS analyzer with an angular resolution of 0.07°. The photon energy was tuned between 350 eV and 650 eV and the combined energy resolution ranged between 50 meV to 100 meV.

Spin-integrated vacuum ultraviolet ARPES (VUV-ARPES) measurements were executed at the high-resolution ARPES branch line of the beamline I05[42] at the Diamond Light Source, United Kingdom. The endstation was equipped with a Scienta R4000 analyzer. Note that in the $k_z$ scan in Fig. 2g) at $h\nu$ above 50 eV, a linear drift of the Gamma point by less than 1° was corrected manually by assuming the spectral

features to be symmetric around Gamma, which they should be due to time-reversal symmetry.

Spin-resolved VUV-ARPES measurements close to the Γ point (Fig. 3(d)i) were carried out at APE-LE endstation at Elettra Synchrotron Trieste, Italy, using a VLEED spin-detector[43]. The MDCs were acquired with alternating magnetization of the spin-filter crystals in a (+$B$, −$B$, −$B$, +$B$) sequence, to average out first-order systematics on the resulting asymmetry spectra.

Spin-resolved surface state ARPES measurements around the R point (Fig. 3(e)ii−vii) were carried out at the COPHEE branch[44] at the SIS endstation of the Swiss Light Source, Switzerland, which is equipped with double classical Mott detectors and has an angle and energy resolution better than 1.5˚ and 75 meV, respectively.

Spin-resolved VUV-ARPES measurements of bulk bands around the R point and the polarization dependence of the Γ-point surface state (Figs. 4 and 5) were performed at the Bloch B endstation at MaxIV, Lund, Sweden, using a Ferrum-VLEED spin-detector with a spin-rotator. Spectra were acquired with alternating magnetization of the spin-filter crystals in a [−++−−++−] sequence, and measurements with different spin rotator settings were carried out subsequently. Infrequent noise from the preamplifier was removed, if a single point deviated by more than 11$\sigma$ from all other points measured with the same settings, as described in detail in Supplementary Note 2.

All ARPES experiments were executed with the sample held around 20 K and at a pressure better than $2 \times 10^{-10}$ mbar. The spin-resolved MDCs in Fig. 3, and Supplementary Figs. [1, 3, 4] were measured perpendicular to the scattering plane and the ones in Fig. 4 within the scattering plane.

For the calculation of the spin polarization, the instrumental asymmetry was corrected by numerically adjusting the detector relative gain between opposite Mott detectors/for opposite spin-filter magnetization. This will affect the absolute value of the polarization, but not its angle dependence. For the analysis of the EDCs in Fig. 5 we assumed a gain of one, since, although mathematically not equivalent, the effect of comparing the polarization to a reference window $E_{ref}$ is almost the same as adjusting a gain, such that there is no polarization within $E_{ref}$. To quantify the orientation of the Fermi-arc spin-polarization, we used a 3D vectorial fit of the measured spin expectation values with a well-established fitting routine[45]. Note that for the s-pol. cut in Fig. 4a position and width of the two surface state peaks were assumed to be the same as in the p-pol. cut from the same figure. For all Fermi arcs, the magnitude of the spin expectation values was assumed to be 1. The Sherman factor has been set 0.3, 0.08, and 0.29 for the data recorded at the APE-LE, COPHEE, and Bloch-B endstations, respectively. When transferring the spin polarization onto the Fermi surface, MDCs measured at the opposite side of the high symmetry point (indicated by negative $k_{\parallel}$-values) were moved to the other side by assuming time-reversal symmetry.

Note that in our measurements of the surface state spin texture, we cannot directly measure the out-of-plane spin component since the signal will be averaged over multiple domains with Fermi arcs spin-polarized in opposite out-of-plane directions. The measured out-of-plane spin texture is therefore always close to zero (see Supplementary Figs. 1, 2, and Supplementary Note 1 for more information).

## Ab-initio calculations

In order to study the spin-polarization properties of the bulk and surface electronic bands we performed density functional theory (DFT) calculations as implemented in the full-potential local-orbital code (FPLO)[46]. We use the Perdew-Burke-Ernzerhof (PBE) parameterization[47] within the generalized gradient approximation (GGA). We used a Monkhorst-Pack grid of $(9 \times 9 \times 9)$ k-points for reciprocal space integration. To analyze the surface spectrum and Fermi arcs we construct an interpolated Hamiltonian in the maximally localized Wannier basis, choosing as starting projections s, p, and d orbitals for Pt and s, p orbitals for Ga. The surface spectrum is computed following the iterative Green's function method as implemented in WannierTools[48].

## Data availability
The primary and processed data generated in this study have been deposited in the Open Research Data Repository of the Max Planck Society under accession code 10.17617/3.GIT9JN [https://doi.org/10.17617/3.GIT9JN], see Ref. 49.

## Code availability
DFT calculations were performed with the FPLO® code available from IFW-Dresden at https://www.fplo.de/. The WannierTools are available at https://github.com/quanshengwu/wannier_tools. ARPES data were analyzed with the MATools package available from https://www.psi.ch/en/sls/adress/manuals, the vectorial fitting routine from Ref. 45 and custom scripts available in the data repository[49].

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

## Acknowledgements

We acknowledge Diamond Light Source for time on beamline I05 under Proposals No. SI20617-1, No. SI24703-1, and No. SI26098, Swiss Light source for beamtime on ADRESS and COPHEE under Proposals No. 20212139, 20210765, 20210484, 20191797, Elettra Sincrotrone Trieste on APE-LE under Proposal No. 20215697, and MAX IV Laboratory on Bloch-B under Proposal No. 20221295. J.A.K. acknowledges support by the Swiss National Science Foundation (SNF-Grant No. P500PT_203159). S.S. acknowledges funding from the Swiss National Science Foundation under project numbers 195133 and 159690. The authors thank Sulamith Gutwein and Theresa Voigt for help with the 3d plots in Fig. 1a, and Oliver J. Clark, Procopi C. Constantinou, Wenxin Li, Jaime Sánchez-Barriga, and Tianlun Yu for help during the ARPES measurements. B.B. acknowledges the support of the U. S. National Science Foundation under grant DMR-1945058. M.G.V. and C.F. thanks support to the Deutsche Forschungsgemeinschaft (DFG, German Research Foundation) - FOR 5249 (QUAST). M.G.V. acknowledges partial support from the European Research Council (ERC) under grant agreement no. 101020833. M.G.V. and I.R. acknowledge that this work has been financially supported by the Ministry for Digital Transformation and of Civil Service of the Spanish Government through the QUANTUM ENIA project call - Quantum Spain project, and by the European Union through the Recovery, Transformation, and Resilience Plan - NextGenerationEU within the framework of the Digital Spain 2026 Agenda. M.G.V. and I.R. acknowledge support from the Spanish Ministerio de Ciencia e Innovacion (PID2022-142008NB-I00). K.M., C.S. and C.F. acknowledge financial support by European Research Council (ERC) Advanced Grant No. 742068 ("TOP-MAT"), Deutsche Forschungsgemeinschaft (DFG) under SFB 1143 (Project No. 247310070) and Würzburg-Dresden Cluster of Excellence on Complexity and Topology in Quantum Matter-ct.qmat (EXC 2147, Project No. 39085490). K.M. acknowledges Max Plank Society for the funding support under Max Plank-India partner group project, and Board of Research in Nuclear Sciences (BRNS) under 58/20/03/2021-BRNS/ 37084/ DAE-YSRA, Science and Engineering Research Board, DST, Government of India, via grant no: CRG/2022/001826 and Aeronautics Research and Development Board (ARDB, Project No. 1992). F.M. gratefully acknowledges the SoE action of PNRR, number SOE_0000068. This work has been partly performed in the framework of the nanoscience foundry and fine analysis (NFFA-MUR Italy Progetti Internazionali) facility. N.B.M.S. was funded by the European Union (ERC Starting Grant ChiralTopMat, project number 101117424). Views and opinions expressed are however those of the author(s) only and do not necessarily reflect those of the European Union or the European Research Council Executive Agency. Neither the European Union nor the granting authority can be held responsible for them.

## Author contributions

J.A.K. and N. B. M. S. designed the experimental plan. J.A.K. and S.S. conducted the ARPES experiments with support from E.C.MF., M.D., B.P., J.Y. and N.B.M.S. The experimental data were analyzed by J.A.K. and

S.S. Ab-initio calculations were performed by I.R. with support from Q.Y. and supervised by M.G.V., whilst M.Li. and B.B. developed the $\mathbf{k} \cdot \mathbf{p}$ model of multifold fermions. The samples were grown by K.M. and characterized and fabricated together with H.B. and C.S. E.B.G., J.H.D., C.M.P., M.Le., V.N.S., M.C., M.D.W., T.K.K., C.C., F.M., J.F. and I.V. maintained the ARPES endstations and provided experimental support and input on data analysis. J.A.K., I.R. and N.B.M.S. wrote the manuscript with input from all co-authors. N.B.M.S., M.G.V., B.B., C.F. and S.S.P.P. supervised parts of the project. N.B.M.S., C.F., B.B. and M.G.V. conceived the idea. N.B.M.S. coordinated the overall project.

## Funding

## Competing interests
The authors declare no competing interest.

## Additional information

[1]Max Planck Institut für Mikrostrukturphysik, Weinberg 2, 06120 Halle, Germany. [2]Department of Physics, University of California, Berkeley, CA, USA. [3]nanotech@surfaces Laboratory, Empa, Swiss Federal Laboratories for Materials Science and Technology, 8600 Dübendorf, Switzerland. [4]Max Planck Institute for Chemical Physics of Solids, Dresden, Germany. [5]Donostia International Physics Center, 20018 Donostia - San Sebastian, Spain. [6]Indian Institute of Technology-Delhi, Hauz Khas, New Delhi 110 016, India. [7]Photon Science Division, Paul Scherrer Institute, 5232 Villigen PSI, Switzerland. [8]Institut de Physique, École Polytechnique Fédérale de Lausanne, 1015 Lausanne, Switzerland. [9]MAX IV Laboratory, Lund University, Fotongatan 2, 22484 Lund, Sweden. [10]Department of Physics, University of Illinois, Urbana-Champaign, USA. [11]Diamond Light Source Ltd, Harwell Science and Innovation Campus, Didcot OX11 0DE, UK. [12]Istituto Officina dei Materiali, Consiglio Nazionale delle Ricerche, Trieste I-34149, Italy. [13]Department of Molecular Sciences and Nanosystems, Ca' Foscari University of Venice, 30172 Venice, Italy. [14]CNR-IOM, Area Science Park, Strada Statale 14 km 163.5, I-34149 Trieste, Italy. [15]Present address: Laboratory for Muon Spin Spectroscopy, Paul Scherrer Institute, CH-5232 Villigen PSI, Switzerland. [16]These authors contributed equally: Jonas A. Krieger, Samuel Stolz, Iñigo Robredo. ✉e-mail: niels.schroeter@mpi-halle.mpg.de

