## [Peer Review File · Nature Communications]

Weyl spin--momentum locking in a chiral topological semimetalReviewers' Comments:

Reviewer #1:

Remarks to the Author:

The manuscript by J.A. Krieger et al. reported the first observation of the spin structure of a chiral topological material PtGa with a Weyl spin-momentum locking (SML). As shown in Fig. 1a, there are three kinds of prototypical forms of SMLs. Two of the Rashba- and Dresselhaus-type SMLs have been investigated so far. However, the last one, Weyl-type SML, has yet to be observed experimentally. In this paper, the authors selected a chiral topological material, PtGa, with a cubic B20 crystal structure. They clarified that the bulk and surface electronic structure with spin directions can be attributed to a Weyl SML using high-energy-resolution spin-resolved SX- and VUV-ARPES.

The authors carefully checked that the spin polarization is not photon polarization-dependent using several synchrotron ARPES beamlines with different setups and obtained consistent results. The careful experiments have yielded unwavering conclusions. Therefore, the referee strongly recommends accepting this manuscript for publication in Nature Communications.

There is a typo as follows:

Page 3, line 3 in the 2nd paragraph: "mutifold" → "multifold"

Reviewer #2:

Remarks to the Author:

Krieger et al performed spin ARPES experiments on a Weyl material PtGa with various photon energies. The authors measured spin polarizations of both surface and bulk states to experimentally prove isotropic spin momentum locking (SML). The isotropic SML is expected to exist only in materials with B20 crystal structure.

The isotropic SML is important because the pseudo-spin becomes the same as real spin when isotropic SML appears. In such case, various physical properties may be obtained from this relation. For example, Chern numbers, usually calculated based on the pseudo spin, can be obtained based on the spin texture. Since isotropic SML only depends on the symmetry, isotropic SML must appear when once a material satisfies the required certain symmetry. Then, isotropic SML can be understood as a symmetry-enforced property, that is, it is very robust.

In fact, the authors provide in-depth discussion on the background/motivation of the research in the introduction. In the introduction, they also discuss what has been done so far, they especially emphasize why their results are different from those in refs 13-16, based on the symmetry argument. In fact, it is known that isotropic SML requires a special symmetry as reported in refs 3 and 10 which materials in refs 13-16 do not have. That is, trigonal tellurium and PtGa belong to different classes of materials and thus the physics associated with PtGa should be considered different. Therefore, I believe the work in this manuscript is quite distinct from the results in refs 13-16.

On the other hand, more important results are those reported in refs 3, 10, 15 and 23 in regards to whether this work presents a substantial fundamental/conceptual advance. Studies on topological materials have been following the standard process of prediction by symmetry analysis/band calculation, followed by experimental verification by ARPES. Symmetry analysis & DFT have worked very well that ARPES results usually play the role at the level of 'confirmation' of the theoretical results. As such, the leading role is often played by the theory. Therefore, whether this work presents a substantial advance or not should be judged based on if this work does something beyond simple confirmation of the calculated spin-dependent electronic structure.

In my view, this work went beyond simple conformation of calculated electronic structure. First of all, this work focuses on the scientific importance of the isotropic SML. Related references of 3, 10, 15, 23 do not really address the issue while predicting parallel Weyl SML. In addition, this work includes

theory part (DFT calculation) which was not done before (as far as I can see). Therefore, it goes beyond simple confirmation. As I mentioned above, isotropic SML is a distinct physics. Counting all these aspects, I believe the manuscript contains results that are novel enough for publication in Nature Communications.

The manuscript is overall well written and contains necessary experimental data to address all the issues. Here are some comments for the authors to consider.

(1) I like the thorough description of what has been done in the introduction. Yet, the introduction is almost 3 pages. A detailed (that is, long) description can sometimes make it difficult to catch the main point. Therefore, the authors may consider reducing it, moving part of it to Supplementary Information.

(2) Authors prepared the sample by sputter & anneal. Are authors sure that the composition on the surface is Pt:Ga=1:1? That is, any possibility for differential sputtering?

(3) What is the inner potential used to determine the k_z value? How was it determined?

(4) In Fig 5, bulk spin-texture of multifold fermion at R is shown. However, spin texture was measured only on the equator of the Fermi sphere at R. Therefore, a true isotropic SML has not been proven with the data in Fig 5. Have authors measured the spin texture at different k_z values?

Reviewer #3:

Remarks to the Author:

The work by J. A. Krieger and coauthors studied the Weyl type spin orbit coupling in the topological semimetal with chiral crystal symmetry. In the past few years, it has been proposed that Weyl points can in principle emerge in any chiral crystals at the time reversal invariant momenta due to the spin orbit coupling. Due to the chiral symmetry, especially the T and O point group, the spin orbit coupling has the form of isotropic $k \cdot \sigma$, so given a Fermi surface enclosing the Weyl point, the spin texture at the Fermi surface shows the monopole like structure, which is known as the Weyl spin orbit coupling or the Weyl spin momentum locking. The Weyl type spin orbit coupling couples the spin orientation of a Bloch state to align parallel (or anti-parallel) to its crystal momentum k , so the chiral crystal with Weyl spin orbit coupling has potential applications in designing the magnetoelectric memory device based on the chiral crystal. In the experiment done here, the authors measured the Weyl spin texture of a chiral crystal PtGa by conducting spin-integrated and spin resolved ARPES measurements. They clearly observed the spin texture of Fermi arcs originate from the Weyl points, which gives the spin texture of bulk bands near the Weyl points indirectly. The spin texture of the bands near the Weyl points are also directly observed.

The Weyl type of spin orbit coupling in a chiral crystal is intrinsically determined by the crystal symmetry and has been clearly studied in theoretical papers. In the bulk response, the longitudinal coupling between the magnetization and the electrons' motion has been reported in experiment in Nat. Commun. 8, 954 (2017), Phys. Rev. Research 3, 023111 (2021). For the spin texture, Ref. 13 and 14 has reported the radial type spin texture of Tellurium as indicated in the draft. However, the surface spin texture of the Fermi arc reported in this work enhances the innovation of the work, so in principle the work can be recommended for the publication. Before that, I will appreciate that the authors can address a few points in the below:

1) The isotropic Weyl type spin orbit coupling, say the form of $k \cdot \sigma$, is lowest linear order approximation. What are the effects of higher order k dependence? From Fig. 1 b and Fig. 5b, it is clear that the spin orientation is not strictly parallel to k and the spin orbit coupling of higher order definitely plays a role. The authors should discuss this point.

2) In this work, the spin orientation of the surface states on Fermi arcs is connected to the Weyl spin orbit coupling in the bulk bands. By measuring the surface spin texture of Fermi arcs, the authors infer the spin texture of the bulk bands. The explicit relation between the spin orbit coupling in the Fermi arc states and the bulk bands near the Weyl points should be given in the main text instead of the supplementary materials.

3) The authors mentioned the interplay between the Weyl spin orbit coupling and the Josephson diode effect. Why does the Weyl spin orbit coupling enabled Josephson diode effect have the advantage of low-dissipation in superconducting memory devices?

REVIEWER COMMENTS

Reviewer #1 (Remarks to the Author):

Reviewer's Comment: The manuscript by J.A. Krieger et al. reported the first observation of the spin structure of a chiral topological material PtGa with a Weyl spin-momentum locking (SML). As shown in Fig. 1a, there are three kinds of prototypical forms of SMLs. Two of the Rashba- and Dresselhaus-type SMLs have been investigated so far. However, the last one, Weyl-type SML, has yet to be observed experimentally. In this paper, the authors selected a chiral topological material, PtGa, with a cubic B20 crystal structure. They clarified that the bulk and surface electronic structure with spin directions can be attributed to a Weyl SML using high-energy-resolution spin-resolved SX- and VUV-ARPES.

The authors carefully checked that the spin polarization is not photon polarization-dependent using several synchrotron ARPES beamlines with different setups and obtained consistent results. The careful experiments have yielded unwavering conclusions. Therefore, the referee strongly recommends accepting this manuscript for publication in Nature Communications.

There is a typo as follows:

Page 3, line 3 in the 2nd paragraph: “mutifold” → “multifold”

Author's reply: We thank the referee for emphasizing the novelty and impact of our work, the high quality of our experimental data, and recommending publication. We have corrected the typo.

Reviewer #2 (Remarks to the Author):

Reviewer's Comment: Krieger et al performed spin ARPES experiments on a Weyl material PtGa with various photon energies. The authors measured spin polarizations of both surface and bulk states to experimentally prove isotropic spin momentum locking (SML). The isotropic SML is expected to exist only in materials with B20 crystal structure.

The isotropic SML is important because the pseudo-spin becomes the same as real spin when isotropic SML appears. In such case, various physical properties may be obtained from this relation. For example, Chern numbers, usually calculated based on the pseudo spin, can be obtained based on the spin texture. Since isotropic SML only depends on the symmetry, isotropic SML must appear when once a material satisfies the required certain symmetry. Then, isotropic SML can be understood as a symmetry-enforced property, that is, it is very robust.

In fact, the authors provide in-depth discussion on the background/motivation of the research in the introduction. In the introduction, they also discuss what has been done so far, they especially emphasize why their results are different from those in refs 13-16, based on the symmetry argument. In fact, it is known that isotropic SML requires a special symmetry as reported in refs 3 and 10 which materials in refs 13-16 do not have. That is, trigonal tellurium and PtGa belong to different classes of materials and thus the physics associated with PtGa should be considered different. Therefore, I believe the work in this manuscript is quite distinct from the results in refs 13-16.

On the other hand, more important results are those reported in refs 3, 10, 15 and 23 in regards to whether this work presents a substantial fundamental/conceptual advance. Studies on topological materials have been following the standard process of prediction by symmetry analysis/band calculation, followed by experimental verification by ARPES. Symmetry analysis & DFT have worked very well that ARPES results usually play the role at the level of ‘confirmation’ of the theoretical results. As such, the leading role is often played by the theory. Therefore, whether this work presents a substantial advance or not should be judged based on if this work does something beyond

simple confirmation of the calculated spin-dependent electronic structure.

In my view, this work went beyond simple conformation of calculated electronic structure. First of all, this work focuses on the scientific importance of the isotropic SML. Related references of 3, 10, 15, 23 do not really address the issue while predicting parallel Weyl SML. In addition, this work includes theory part (DFT calculation) which was not done before (as far as I can see). Therefore, it goes beyond simple confirmation. As I mentioned above, isotropic SML is a distinct physics. Counting all these aspect, I believe the manuscript contains results that are novel enough for publication in Nature Communications.

Author's reply: We thank the referee for highlighting the novelty of our work and considering our results worthy of publication in Nature Communications.

Reviewer's Comment: The manuscript is overall well written and contain necessary experimental data to address all the issues. Here are some comments for the authors to consider.

(1) I like the thorough description of what has been done in the introduction. Yet, the introduction is alost 3 pages. A detailed (that is, long) description can sometimes make it difficult to catch the main point. Therefore, the authors may consider reducing it, moving part of it to Supplementary Information.

Author's reply: As per the referees' suggestion, we have shortened the introduction slightly.

Reviewer's Comment: (2) Authors prepared the sample by sputter & anneal. Are authors sure that the composition on the surface is Pt:Ga=1:1? That is, any possibility for differential sputtering?

Author's reply: Whilst differential sputtering is generally a possibility in binary compounds. However, we believe that after annealing, PtGa displays a surface stoichiometry that is very similar to the bulk. The sputter-annealing recipe was originally developed for the iso-structural compound PdGa, where careful investigations with STM and angle resolved XPS found no indications of elemental segregation [Rosenthal, et al., Langmuir 28 (17), 6848-6856 (2012), <https://doi.org/10.1021/la2050509>]. We note that both in PtGa and PdGa, the unreconstructed LEED pattern with low background serves as a good indication of a bulk-truncated surface. Moreover, the high-quality ARPES spectra indicate a low-defect concentration at the surface.

In addition, we can roughly crosscheck that the Pt/Ga ratio is reasonable based on XPS survey spectra measured at a photon energy of $h\nu=1050\text{eV}$ during the SX-ARPES experiment:

Fig. R1: (a) Overview XPS spectra of PtGa 111. (b,c) The Pt4f and Ga3d peaks with the solid lines showing the fits used to estimate the atomic ratio.

To estimate the atomic ratio we normalize the peak areas by the corresponding calculated atomic photoionization cross-section, see <https://vuo.elettra.eu/services/elements/WebElements.html> and [J.J. Yeh, Atomic Calculation of Photoionization Cross-Sections and Asymmetry Parameters, Gordon and Breach Science Publishers, Langhorne, PE (USA), 1993]. For the data shown above, this gives a ratio of Pt/Ga~0.98, confirming the expected order of magnitude of the atomic ratios of the bulk stoichiometry.

We have included this XPS analysis into the revised supplementary materials as a new section S10.

Reviewer's Comment: (3) What is the inner potential used to determine the k_z value? How was it determined?

Author's reply: The photon energies of the high symmetry points were determined experimentally via photon energy dependent ARPES measurements. Matching those scans to the expected Brillouin zone periodicity results in an inner potential $V_0=12\text{eV}$ (Fig. R2).

Fig. R2: Out-of-plane k_z Fermi surface of PtGa 111 adapted from [Yen, et al., arxiv:2311.13217 (2023), <https://doi.org/10.48550/arXiv.2311.13217>].

At lower photon energies it is more difficult to experimentally identify the high symmetry points due to the significant increase of k_z -broadening. However the spectra are still consistent with $V_0=12\text{eV}$. This is shown in Fig. R3, which is the spectrum from Fig. 2(g) with overlaid Brillouin zone boundaries. Moreover, the similarity between the bulk-sensitive SX-ARPES measurements shown in Fig. 2a and the spectrum measured at $h\nu=74\text{ eV}$ (Fig. 5a) further supports the correct choice of photon energies.

We have included a statement about the inner potential and correct choice of photon energies into the main text.

Fig. R3: Out-of-plane k_z Fermi surface of PtGa 001 (Fig.2g) with Brillouin zone boundaries. The green line shows the constant $h\nu$ cut corresponding to Fig. 5 in the manuscript.

Reviewer's Comment: (4) In Fig 5, bulk spin-texture of multifold fermion at R is shown. However, spin texture was measured only on the equator of the Fermi sphere at R. Therefore, a true isotropic SML has not been proven with the data in Fig 5. Have authors measured the spin texture at different k_z values?

Author's reply: We have not measured at other photon energies with spin-resolution because these measurements are extremely time-consuming due to the low efficiency of the spin-detectors. Moreover, synchrotron beamtime at specialized spin-ARPES beamlines is scarce. However, the cubic symmetry of the system, and the three fold rotation axis around the 111 direction, the in-plane k_x , k_y directions are symmetry equivalent to the k_z direction. This results in the following sampling of the 3D Fermi surface:

Fig. R4: Symmetry equivalent points (red) to the ones measured in Fig. 5. (The Gamma point Fermi surface has been overlaid for reference.)

We agree with the referee that it would be of interest to also measure at an off-symmetry point where all of k_x , k_y , and k_z are nonzero. Note, however, that k_z broadening is not a big problem at the equator, because the Fermi surface changes only slowly along the k_z direction. We, therefore, believe that this measurement would most likely require a higher k_z resolution than what is currently feasible with existing VUV-spin-ARPES beamlines. Whilst access to spin-resolved soft X-ray ARPES is currently extremely limited, there are promising instrumental developments underway (e.g. at the Swiss Light Source and ALBA synchrotron) that will hopefully make these experiments feasible in the near future.

Reviewer #3 (Remarks to the Author):

Reviewer's Comment: The work by J. A. Krieger and coauthors studied the Weyl type spin orbit

coupling in the topological semimetal with chiral crystal symmetry. In the past few years, it has been proposed that Weyl points can in principle emerge in any chiral crystals at the time reversal invariant momenta due to the spin orbit coupling. Due to the chiral symmetry, especially the T and O point group, the spin orbit coupling has the form of isotropic $\mathbf{k} \cdot \boldsymbol{\sigma}$, so given a Fermi surface enclosing the Weyl point, the spin texture at the Fermi surface shows the monopole like structure, which is known as the Weyl spin orbit coupling or the Weyl spin momentum locking. The Weyl type spin orbit coupling couples the spin orientation of a Bloch state to align parallel (or anti-parallel) to its crystal momentum \mathbf{k} , so the chiral crystal with Weyl spin orbit coupling has potential applications in designing the magnetoelectric memory device based on the chiral crystal. In the experiment done here, the authors measured the Weyl spin texture of a chiral crystal PtGa by conducting spin-integrated and spin resolved ARPES measurements. They clearly observed the spin texture of Fermi arcs originate from the Weyl points, which gives the spin texture of bulk bands near the Weyl points indirectly. The spin texture of the bands near the Weyl points are also directly observed.

The Weyl type of spin orbit coupling in a chiral crystal is intrinsically determined by the crystal symmetry and has been clearly studied in theoretical papers. In the bulk response, the longitudinal coupling between the magnetization and the electrons' motion has been reported in experiment in Nat. Commun. 8, 954 (2017), Phys. Rev. Research 3, 023111 (2021). For the spin texture, Ref. 13 and 14 has reported the radial type spin texture of Tellurium as indicated in the draft. However, the surface spin texture of the Fermi arc reported in this work enhances the innovation of the work, so in principle the work can be recommended for the publication.

Author's reply: We thank the referee for recognizing the novelty of our work and considering it worthy of publication in Nature Communications. We have included the two mentioned references into our paper. As an aside, as we point out in the introduction section of the manuscript, cubic materials like PtGa can show *isotropic* parallel spin-momentum locking, whilst materials of lower symmetry such as trigonal tellurium actually only show parallel spin-momentum locking along a single high symmetry axis, but not along any generic direction.

Reviewer's Comment: Before that, I will appreciate that the authors can address a few points in the below:

1) The isotropic Weyl type spin orbit coupling, say the form of $\mathbf{k} \cdot \boldsymbol{\sigma}$, is lowest linear order approximation. What are the effects of higher order \mathbf{k} dependence? From Fig. 1 b and Fig. 5b, it is clear that the spin orientation is not strictly parallel to \mathbf{k} and the spin orbit coupling of higher order definitely plays a role. The authors should discuss this point.

Author's reply: We thank the referee for this suggestion. To clarify this point, we have now included the full Hamiltonian from the supplementary S19 in the main text, which contains all the terms of spin-orbit coupling in linear order. This includes the material dependent SOC parameters λ_i .

Depending on the momentum \mathbf{k} and the λ_i , there can be three regimes of spin-momentum locking (see supplementary section S7 for details). We have now performed additional numerical fits of the $\mathbf{k} \cdot \mathbf{p}$ model to the DFT calculations which show that for the states near the Fermi level at the Gamma point, we are in the regime $\lambda_i |\mathbf{k}|$ ($i=1,2,3$) $< \lambda_0 < v_F |\mathbf{k}|$, where v_F is the Fermi velocity and λ_i ($i=1,2,3$) are coefficients for SOC corrections linear in \mathbf{k} , whilst λ_0 is the SOC correction constant in \mathbf{k} .

In this regime, one may neglect the contributions from λ_i and would then find perfect parallel spin-momentum locking to lowest order in perturbation theory. As the referee correctly points out, there are corrections visible in the DFT calculations. These corrections originate from:

- 1) linear-in-k corrections that depend on the terms $\lambda_i|k|$. At the Fermi-level, the largest contribution is from $\lambda_3|k| \sim 10$ meV, which is much smaller than $\lambda_0 \sim 100$ meV. Thus these linear in k corrections are expected to be small.
- 2) On the same order of magnitude, there are second order contributions proportional to λ_0^2
- 3) Finally, there are also second order corrections in momentum (k) to the SOC-free Hamiltonian

In summary, when going beyond first order in perturbation theory, the model will quickly grow with many additional parameters. However, from the DFT calculations, we find that for PtGa near the Fermi level, the total effect of these corrections is relatively small: For the inner Fermi-surface pocket near the Γ point we find angular deviations on the order of $\sim 1^\circ$, and up to around $\sim 30^\circ$ for selected points on the outer Fermi-surface where the Fermi-momentum is larger and therefore linear-in-k and quadratic-in-k corrections become more significant.

We have extended the discussion of the Hamiltonian in the main text below equation 1, and also included the discussion of the corrections in the revised supplementary section S7.

Reviewer's Comment: 2) In this work, the spin orientation of the surface states on Fermi arcs is connected to the Weyl spin orbit coupling in the bulk bands. By measuring the surface spin texture of Fermi arcs, the authors infer the spin texture of the bulk bands. The explicit relation between the spin orbit coupling in the Fermi arc states and the bulk bands near the Weyl points should be given in the main text instead of the supplementary materials.

Author's reply: We have extended the Hamiltonian in the main text of Eq. 1 to the full Hamiltonian containing all the symmetry allowed spin-orbit coupling terms given in the supplementary materials Eq. S19, and also updated the subsequent main text to describe the effect of spin-orbit coupling on the Fermi-arc surface states. However, we believe that moving the full derivation of the theoretical Fermi-arc model from the supplementary section 7 into the main text would make the the corresponding section of the text too long (see also comment 1 of referee #2 asking us to shorten this section) and thus limit the readability of our manuscript.

Reviewer's Comment: 3) The authors mentioned the interplay between the Weyl spin orbit coupling and the Josephson diode effect. Why does the Weyl spin orbit coupling enabled Josephson diode effect have the advantage of low-dissipation in superconducting memory devices?

Author's reply: The Josephson diode effect (independent of its origin) could be used to read out a magnetic state or domain wall in a superconducting memory devices, see e.g., [Hess, et al., PRB 108, 174516 (2023), <https://doi.org/10.1103/PhysRevB.108.174516>]. Such a read-out would be low dissipation, because depending on the magnetic state, there would be a dissipationless supercurrent flowing through the junction. The advantage of the Weyl-type spin orbit coupling materials is that it allows for device geometries that are not possible with existing materials with Rashba-type SOC, as we discuss in our supplementary materials section S9.

We have included the reference [Hess, et al., PRB 108, 174516 (2023), <https://doi.org/10.1103/PhysRevB.108.174516>] into the main text.

Reviewers' Comments:

Reviewer #2:

Remarks to the Author:

In the resubmitted version of the manuscript, Krieger et al provided appropriate answers to my questions and comments. Here are my final comments for the reviewers to consider.

(1) OK

(2) Authors' arguments against differential sputtering is very reasonable. I agree with the authors.

(3) It is good that the authors added the information to the manuscript.

(4) Authors are right that k_z broadening effect should be negligible near the equator. In addition, the cubic symmetry allows us to use the data from an equator for other equators as the authors pointed out. I suggest authors briefly mention these points in the manuscript.

Reviewer #3:

Remarks to the Author:

The authors now have well addressed the points raised before, so now I can recommend the publication of this work.

REVIEWER COMMENTS

Reviewer #2 (Remarks to the Author):

Reviewer's Comment: In the resubmitted version of the manuscript, Krieger et al provided appropriate answers to my questions and comments. Here are my final comments for the reviewers to consider.

- (1) OK
- (2) Authors' arguments against differential sputtering is very reasonable. I agree with the authors.
- (3) It is good that the authors added the information to the manuscript.
- (4) Authors are right that k_z broadening effect should be negligible near the equator. In addition, the cubic symmetry allows us to use the data from an equator for other equators as the authors pointed out. I suggest authors briefly mention these points in the manuscript.

Author's reply: We thank the referee for their positive recommendation. In response to point (4) we have added the following text to the manuscript:

To measure the spin-polarization of this band we have acquired energy distribution curves (EDCs) at different **locations** around the R point **in the $kz=0$ plane, where the kz broadening is reduced because the Fermi surface changes only slowly along the kz direction. Due to the crystal symmetry the obtained results are representative for equivalent points in the $kx=0$ and $ky=0$ planes as well. The positions of these cuts are indicated with Roman numerals on a constant energy surface** in Fig. 5b, which also shows the calculated spin texture from ab-initio models. Here the spin-direction is indicated by yellow arrows and shows a Weyl-type parallel spin--momentum locking. To investigate potential matrix element effects, we obtained spin-resolved measurements with four different light polarizations (Fig. 5c).

Reviewer #3 (Remarks to the Author):

Reviewer's Comment: The authors now have well addressed the points raised before, so now I can recommend the publication of this work.

Author's reply: We thank the referee for their positive recommendation.